# Supplementation of a High-Fat Diet with Pentadecylresorcinol Increases the Representation of *Akkermansia muciniphila* in the Mouse Small and Large Intestines and May Protect against Complications Caused by Imbalanced Nutrition

**DOI:** 10.3390/ijms25126611

**Published:** 2024-06-15

**Authors:** Anastasia A. Zabolotneva, Ilya Yu. Vasiliev, Tatiana Grigoryeva, Andrei M. Gaponov, Vladimir P. Chekhonin, Sergei A. Roumiantsev, Aleksandr V. Shestopalov

**Affiliations:** 1Department of Biochemistry and Molecular Biology, Faculty of Medicine, N. I. Pirogov Russian National Research Medical University, 1 Ostrovitianov Str., Moscow 117997, Russia; chekhoninnew@yandex.ru (V.P.C.); rumyantsev.sergey@endocrincentr.ru (S.A.R.); aleksandr.shestopalov@fccho-moscow.ru (A.V.S.); 2Laboratory of Biochemistry of Signaling Pathways, Endocrinology Research Center, 11 Dm. Ulyanova Str., Moscow 117036, Russia; mepk_m6@mail.ru; 3Institute of Fundamental Medicine and Biology, Kazan Federal University, 18 Kremlyovskaya Street, Kazan 420008, Russia; tatabio@inbox.ru; 4V. A. Negovsky Research Institute of General Reanimatology, Federal Research and Clinical Center of Intensive Care Medicine and Rehabilitology, Moscow 141534, Russia

**Keywords:** alkylresorcinol, pentadecylresorcinol, high-fat diet, small intestinal microbiota, large intestinal microbiota, *Akkermansia muciniphila*, *Bifidobacterium pseudolongum*

## Abstract

Imbalanced nutrition, such as a high-fat/high-carbohydrate diet, is associated with negative effects on human health. The composition and metabolic activity of the human gut microbiota are closely related to the type of diet and have been shown to change significantly in response to changes in food content and food supplement administration. Alkylresorcinols (ARs) are lipophilic molecules that have been found to improve lipid metabolism and glycemic control and decrease systemic inflammation. Furthermore, alkylresorcinol intake is associated with changes in intestinal microbiota metabolic activity. However, the exact mechanism through which alkylresorcinols modulate microbiota activity and host metabolism has not been determined. In this study, alterations in the small intestinal microbiota (SIM) and the large intestinal microbiota (LIM) were investigated in mice fed a high-fat diet with or without pentadecylresorcinol (C15) supplementation. High-throughput sequencing was applied for jejunal and colonic microbiota analysis. The results revealed that C15 supplementation in combination with a high-fat diet could decrease blood glucose levels. High-throughput sequencing analysis indicated that C15 intake significantly increased (*p* < 0.0001) the abundance of the probiotic bacteria *Akkermansia muciniphila* and *Bifidobacterium pseudolongum* in both the small and large intestines and increased the alpha diversity of LIM (*p* < 0.05), but not SIM. The preliminary results suggested that one of the mechanisms of the protective effects of alkylresorcinol on a high-fat diet is the modulation of the content of SIM and LIM and metabolic activity to increase the probiotic bacteria that alleviate unhealthy metabolic changes in the host.

## 1. Introduction

A prolonged unhealthy diet, such as a Western diet characterized by high consumption of saturated fats and refined sugars, is often associated with the development of insulin resistance and impaired glucose tolerance, low-grade systemic inflammation, dyslipidemia, dysbiosis, and endotoxemia [1]. These pathological metabolic changes are associated with different metabolic disorders, such as obesity, type 2 diabetes mellitus, cardiovascular disease, and cancer [2]. Furthermore, a high-fat diet (HFD) significantly changes the composition and function of the intestinal microbiota [3]. In the study by Hildebrandt et al. [4], it was shown that a high-fat diet itself, regardless of obesity status or lean state, caused changes in the intestinal microbiota of mice. Experiments on intestinal microbiota transplantation from HFD-fed mice to germ-free mice have also confirmed the appearance of diet-induced dysbiosis [5,6,7]. On the other hand, a diet can contain ‘metabolic protectors’, positively modulating the gut microbiota community and preventing the development of metabolic disorders [8,9,10,11,12]. For example, alkylresorcinols (ARs) are polyphenolic lipids synthesized primarily by cereals [13] but also by bacteria and fungi [14]. As a component of grain kernels, ARs are believed to determine the health-protective properties of whole grain products [15]. Liu et al. [16] showed that dietary AR supplementation significantly improved glucose tolerance and restored serum glucagon-like peptide 1 (GLP-1) levels in C57BL/6J mice fed HFD compared to those of the control group. In the study by Reshma et al. [17], the effects of olivetol (pentylresorcinol) on a diet-induced obese zebrafish model were investigated. Researchers have shown that pentylresorcinol reduces excessive fat accumulation and triglyceride and lipid accumulation, thus exerting an antiobesity effect and demonstrating therapeutic potential for the treatment of hyperlipidemia-related disorders [17]. Furthermore, 5-heptadecylresorcinol has been found to improve HFD-induced mitochondrial dysregulation in mouse skeletal muscle tissue [18].

These effects of ARs may be caused by the modulatory effect of ARs on the gut microbiota, as shown in previous studies [16] and in our previous work [19]. However, in these works, the influence of AR was taken into account only in relation to the large intestinal microbiota (LIM) or solely to the fecal microbiota; although, the host’s metabolic, immunological, and endocrine functions, among other physiological processes, may be greatly affected by the small intestine microbiota (SIM). Studies have also demonstrated the importance of the small intestinal microbiota in causing alterations in the bacterial population in the small intestine and intestinal abnormalities in mice and rats, as well as serving as dietary signal transducers [20].

5-Pentadecylresorcinol (C15) is an AR with a side chain of 15C length, which potentially has health-protective properties. In our study, a 4-week high-fat diet with or without C15 supplementation was used to establish a feeding mouse model and high-throughput sequencing was used to investigate changes in the microbiota composition and metabolic activity in the small intestine of mice compared to those in the large intestine of the same mice. Additionally, a preliminary discussion was held on the function of dietary AR supplements, SIM and LIM, and imbalanced nutrition in light of the collected data.

## 2. Results

### 2.1. Body Weight (BW) and Food and Water Intake

Before the start of the experiment, the average BW of the mice for all groups was 14.4 ± 0.96 g (BW varied in different groups no more than 10% of average weight). After 4 weeks of feeding, the BW was not markedly different between the six groups (Figure 1a). The average increase in BW in the experimental groups of mice was 0.94 g/week. Food intake in all groups was 2.3 ± 0.18 g/day.

### 2.2. Fasting Blood Glucose (FBG), Serum Triacylglyceride (TAG), and Cholesterol Levels

FBG levels were comparatively higher in the HFD group than in the SD (*p* < 0.0001) group (*p* < 0.0001) and in the group (*p* < 0.0001 for the SD + C15 and *p* < 0.01 for the HFD + C15 groups; Figure 1b). However, serum levels of TAG and cholesterol did not differ between groups. The possible reason for the absence of alterations in lipid metabolism may be a short-term period of feeding, which was not enough to cause perturbations in lipid balance.

### 2.3. Composition of the Gut Microbiota in the Mouse Small and Large Intestines

The diversity, richness, and composition of the jejunum and colon microbiota were analyzed in pairs of mice fed a standard or a high-fat diet with or without C15 supplementation, respectively. A total of 14,117,726 effective 16S rRNA raw reads were obtained from mouse jejunum and colon microbiota samples. We identified more than 10,000 distinct OTUs that occurred at least once for each sample in the SI and LI (Appendix A). However, 99 distinct OTUs were detected everywhere in each sample (Appendix A), of which 81 were unique to the LI, and 5 were unique to the SI.

Alpha diversity was used to express richness, diversity, and evenness (Table 1). The Chao 1 richness index and the different OTUs (observable OTU number) indicated that the colon microbiota of the HFD + C15 feeding group was significantly more abundant than that of the control group (*p* < 0.05; Figure 2a,b). However, no significant differences were observed in the jejunum microbiota, and there were no differences in diversity or evenness between the experimental groups (Table 1). These results illustrated that C15 intake in combination with a high-fat diet increased the richness of the colon microbiome.

Principal coordinate analysis (PCoA) based on unweighted UniFrac distances for SIM and LIM was performed for standard diet (Figure 3a) and high-fat diet (Figure 3b) samples to determine the distribution of different samples according to the structure of the bacterial community.

ANOSIM of the Bray-Curtis distances revealed that the magnitude of differences in the bacterial communities of SIM and LIM is small in any of the study groups (*R* = 0.2527, *p* = 0.001) (Appendix A). These results indicated that the diversity within the SIM or LIM community was more obvious than between them for all study groups.

Therefore, analysis of the SIM and LIM beta diversity did not reveal significant differences between the bacterial communities across all the study groups. Additional indices that evaluate the beta diversity of the microbial communities of SI and LI are presented in Appendix A.

### 2.4. Composition of the Microbial Community in the Mouse Small and Large Intestines

To investigate the effects of a high-fat diet combined with C15 supplementation on the SIM and LIM compositions of mice, we carried out a metagenomic sequencing analysis of the jejunum and colon microbiota of all mice in six different groups (except for group 5 (see Table 1), which was fed a standard diet supplemented with C15, in which we were able to obtain data for 11 samples only). At the phylum level, the microorganisms belonged primarily to seven phyla: Firmicutes, Tenericutes, Bacteroidota, Proteobacteria, Verrucomicrobia, Actinobacteria and TM7. Firmicutes was the most dominant phylum in all the groups (Figure 4a,d). However, the Firmicutes/Bacteroidota ratio did not change under the different diet conditions.

According to Figure 4e,f, the abundance of the Actinobacteria and Verrucomicrobia phyla increased dramatically in the SIM of the HFD + C15 group compared to that of the other experimental and control groups. We also observed a significant increase in the Verrucomicrobia phylum in LIM (Figure 4c) in mice fed an HFD + C15, while the abundance of Bacteroidota decreased in the same group (Figure 4b).

Appendix A and Figure 5a–e show the changes in the levels of class, order, family, genus, and species of bacteria. We observed a marked increase in the representation of the Coriobacteriia and Verrucomicrobiae classes in the small intestine in the HFD + C15 group compared to that in the HFD or SD group (Appendix A), while differences in the representation of the Mollicutes, Eysipelotrichi and Bacilli classes were apparently related to the influence of the solvent (Appendix A). Concerning the large intestine, we observed a significant increase in the abundance of the Actinobacteria and Verrucomicrobiae classes (Appendix A), whereas the abundance of the Bacteroidia and Bacilli classes decreased (Appendix A) in the mice fed with HFD + C15. However, we did not observe a difference in the Bacillus content between HFD + EtOH- and HFD + C15-fed mice, which points to the predominant influence of ethanol on the representation of microbes. These observations were also confirmed by the following analysis at the order (Appendix A), family (Appendix A) and genus (Appendix A) levels.

Importantly, there was a significant increase in the representation of *Akkermansia muciniphila* and *Bifidobacterium* sp. (including *B. pseudolongum*) in both the small and the large intestines of mice fed a high-fat diet with C15 supplementation compared to those in the other study groups (Figure 5a–e). However, in the small intestine, a difference in *A. muciniphila* representation was confirmed between the groups fed a regular diet and the HFD + C15 diet (*p* < 0.01) (Figure 5c).

### 2.5. Reconstruction of the Metabolic Activity of the Mouse Gut Microbiota

To establish whether C15 supplementation causes changes in intestinal microbiota metabolic activity, we performed a PICRUSt2 analysis based on metagenome sequencing data, which allowed us to estimate the representation of bacterial genes in the SIM and LIM communities.

According to the results of PICRUSt2, we revealed some differences in the metabolic activity of the SIM community between mice fed an SD and those fed an SD + C15 diet (Appendix A) or an HFD + C15 diet (Table 2; Appendix A). Namely, the representation of methanogenesis via the acetate pathway decreases after C15 supplementation. This pathway is related to anaerobic Archaea methanogens [21]. The abundance of H_2_-producing methanogens in the gut microbial community is believed to be associated with obesity and its complications in the host [22]. On the other hand, the abundance of probiotic bacteria, such as Bifidobacterium shunt or peptidoglycan biosynthesis (*Enterococcus faecium*), was significantly increased in the SIM of mice that received C15 along with a HFD but not in those that received SD (Table 2).

In the SIM of HFD or SD fed mice compared to HFD + C15 mice, the number of pathways associated with quinol biosynthesis (such as pathways for ubiquinol-8 or naphthoate biosynthesis) was also increased after C15 supplementation (Table 2 and Table 3).

Analysis of the metabolic activity of mouse LIM communities showed that supplementation of HFDs with C15 increased the representation of the quinol synthesis pathways (Table 4); although, there were no differences in the representation of bacterial genes under standard diet conditions.

## 3. Discussion

Current Western societies are characterized by a trend towards unbalanced nutrition. Quick snacks and fast foods enriched with simple sugars, fats, and flavorings create the impression of satiety and food satisfaction. Although there is an assurance that only a prolonged high-fat/high-carbohydrate diet leading to obesity and other metabolic diseases can cause sustainable changes in host and gut microbiota metabolism, in recent years, additional evidence of the crucial influence of a short-term high-fat diet on the gut microbiota and host metabolism has appeared [17,23,24,25]. It is known that diet-induced changes in the gut microbiome occur rapidly (within days) [26].

In the present investigation, we showed that a 4-week high-fat diet led to significantly higher fasting blood glucose levels in mice than a regular diet (Figure 1b), indicating the development of impaired glucose tolerance. On the other hand, we did not observe disturbances in lipid metabolism, possibly due to the lack of a prolonged dietary period.

Resorcinolic phenols are natural compounds with a wide range of biological effects. For example, antioxidant [27], anti-inflammatory [28] and anticarcinogenic [29] effects have been demonstrated for alkylresorcinols with different side chain lengths [27]. In a recent study by Reshma et al., a short-term high-fat diet-induced obese zebrafish model was investigated, and pentylresorcinol was shown to protect against obesity by preventing excess fat accumulation [17]. In a mouse model of obesity induced by a high-fat diet, compared to an HFD, AR supplementation was also shown to significantly improve glucose homeostasis and restore the serum level of GLP-1 [16] and increase glucose tolerance and insulin sensitivity by suppressing liver lipid accumulation and intestinal cholesterol absorption [30]. Furthermore, AR treatment was found to significantly improve the abundance of short-chain fatty acid-producing bacteria, such as *Bacteroides*, *Bifidobacterium* and *Akkermansia* [16]. All of these observations are in agreement with the results of our study, in which we demonstrated that pentadecylresorcinol supplementation in combination with a high-fat diet decreases blood glucose levels and markedly increases the abundance of *Bifidobacterium pseudolongum* and *Akkermansia muciniphila* in both the small and large intestine of mice.

For the first time, we studied the influence of C15 supplementation on mouse SIM and LIM composition and metabolic activity. We found that although a short-term high-fat diet with or without C15 addition did not change the beta diversity of SIM or LIM, the HFD + C15 diet increased the alpha diversity of the LIM community compared to that of the SD or HFD groups. Furthermore, C15 supplementation changed the metabolic activity of SIM and LIM, as predicted by the PICRUSt2 algorithm. Analysis of differentially represented metabolic pathways among more than 400 microbial pathways has shown that a diet supplemented with C15 leads to a decrease in the abundance of methanogenic pathways in SIM but a greater representation of pathways associated with quinol synthesis (such as biosynthesis of menaquinols and naphthanoate derivative biosynthesis) in both SIM and LIM.

Increased methanogenesis is known to be associated with obesogenic microbiota in mice and humans [31]. There are several factors that induce the activity of methanogens in the intestine, such as a strictly anaerobic environment and excess energy-rich sources supporting the growth of methanogens. Methanogens can promote the increased fermentation of polysaccharides and the production of nutrients by nearby microbes, which can lead to weight gain in the host. Methane gas also alters intestinal motility and slows intestinal transit, which can prolong the amount of time required for nutrients to be absorbed [32]. Previously, we found that the gut microbiota can potentially synthesize C15 [19]. Furthermore, we showed that, among the short- and medium-chain ARs investigated, C15 had the highest concentration in the stool of obese children, and the stool content of C15 was negatively correlated with the expression of genes that encode enzymes involved in methane synthesis in methane-producing microorganisms (*p* < 0.05); moreover, no similar statistically significant correlation was found between these two parameters in healthy children [33]. Therefore, C15 supplementation could be one of the regulatory factors for the representation of methanogens in the intestine. Furthermore, we found that C15 increased the abundance of metabolic pathways specific for probiotic bacteria, such as the Bifidobacterium shunt and peptidoglycan biosynthesis IV (*Enterococcus faecium*), which may indicate the activation of metabolic activity in these microbes. It is well known that *Bifidobacterium* sp., *E. faecium,* and *A. muciniphila* are residents of the gut microbiota and are beneficial to host health. Administration of *E. faecium* to HFD-fed mice was shown to result in markedly decreased weight gain and led to increased abundance of *Bifidobacterium* and *A. muciniphila* in the mouse intestine [34]. *Bifidobacterium sp*. and *A. miciniphila* are also used as probiotics due to their ability to improve intestinal dysbiosis, recover the mucus barrier, improve nutrient digestion, and produce biologically active compounds such as SCFAs important for intestinal function and support of host metabolism [35,36,37]. Therefore, the ability of C15 to increase the abundance of probiotic species and their metabolic activity is very promising in the context of treating metabolic dysfunction and supplementing food that allows reducing the negative effects of a high-fat or imbalanced diet.

In HFD-fed mice, the administration of C15 also caused an increase in the abundance of the quinol (coenzyme Q and menaquinone) biosynthesis pathway in SIM and LIM. On the one hand, this observation could be related to an overall increase in the metabolic activity of the gut microbiota because quinol biosynthesis is a highly conserved pathway represented by all aerobic and anaerobic bacteria and is essential for supplying energy to the cell. However, quinols possess various functions in bacterial systems, such as regulation of gene expression, sporulation, and bacterial virulence. Thus, the effects of C15 could be complex due to the signaling functions of AR. However, these mechanisms are still challenging topics in the study of AR effects.

The diverse bacterial communities found in various parts of the human intestine are essential for human health [36]. Due to their relatively lower accessibility, lower abundance, and anaerobic environment, studies on small intestine symbionts have been insufficient compared to the number of fecal microorganisms represented by the microbiota of the large intestine. In our study, for the first time, we have shown the impact of AR supplementation on the composition of the jejunum and colon microbiota and metabolic activity under conditions of a short-term high-fat diet. These findings showed that pentadecylresorcinol can be considered a means of treating dysbiosis and improving the metabolic function of the microbiota through stimulation of probiotic bacterial growth in both the small and large intestines.

## 4. Materials and Methods

### 4.1. Experimental Animals and Study Design

All experimental animal procedures were approved by the Ethics Committee for Animal Research, I.M. Sechenov First Moscow State Medical University, Moscow, Russia (protocol number 96 from 2 September 2021). All experimental procedures were performed according to the relevant guidelines and regulations. All methods are reported following the ARRIVE guidelines.

C57BL/6SPF mice (*n* = 72, females) (reared at the Laboratory Animal Nursery in Puschino, Puschino, Russia) were housed in the animal center of the SPF level of Sechenov First Moscow State Medical University (Moscow, Russia) under the following conditions: 22 °C, 55% humidity, and 12 h: 12 h light–dark cycle. The experimental animals were given ad libitum access to sterile food (Altromin 1324 FORTI, Lage, Germany) and water one week before the official trial began. Following the adaption period, the mice were randomly assigned to six groups of twelve animals each, the groups being separated by no more than ±10% differences of the total weight. Mice were 4–5 weeks old and weighed 14.4 ± 0.96 g on average at the start of the experiment. A high-fat dietary model was generated by feeding laboratory animals a high-fat diet enriched with animal-derived triglycerides and providing up to 30% of total calories (Altromin C 1090-30, Lage, Germany). The food included 13.3% crude fat, 21.1% crude protein, 5.1% crude fiber, 3.9% crude ash, 50.8% nitrogen-free extractives and 5.8% moisture starting from the age of 4–5 weeks until the end of the experiment. The animals in the control group were fed a regular food diet (Altromin 1324 FORTI, Lage, Germany) throughout the investigation period. 5-n-Pentadecylresorcinol (C15) (Hangzhou ROYAL Import & Export Co., Ltd., Hangzhou, China) was administered along with a standard or a high-fat diet through an atraumatic intragastric tube at a dose of 0.006 mg/day per mouse for 28 days. To improve the solubility of C15, ethanol was added to the solution at a concentration of 0.8% *w*/*w*. Therefore, the additional control groups of the animals received 0.2 mL of water solution containing 0.016 g of ethanol per day in the same manner as the high-fat or standard diet. The mice were sacrificed after 28 days of feeding after anesthetization with isoflurane (RWD Life Science, Chenzhen, China).

### 4.2. Body Weight (BW), Food and Water Intake, Fasting Blood Glucose (FBG), Serum Triglyceride (TG), and Cholesterol Levels Were Measured

BW, food, and water intake were recorded weekly. Serum levels of FBG, TG, and cholesterol were measured to assess the effects of C15 on systemic metabolism. The animals were fasted for 5 h with free access to water before the FBG test. Blood samples were taken from anesthetized mice by the eyeball enucleation method, kept at room temperature for 2 h and centrifuged (1500 rpm at 4 °C) for 30 min to separate serum. Glucose, TG and cholesterol concentrations were determined using GLUC3, TRIGL and CHOL2 test kits, respectively (Roche, Basel, Switzerland; catalog numbers: 04404483190, 05171407188, and 03039773190, respectively), and a Cobas c311 blood analyzer was used (Roche, Switzerland).

### 4.3. Large Intestinal and Small Intestinal Microbiota Sampling for Metagenome Analysis

The jejunum and colon tissues were taken sterile [38]. We considered that the duodenum is the first 4 cm of the stomach, beginning with the pylorus. After disposal of 2 cm of the small intestine between the duodenum and jejunum, 6 cm of jejunum was collected, snap frozen in liquid nitrogen and then kept at −80 °C until analysis. The jejunum and its contents were sterilely sectioned into 1 cm long pieces for metagenomic analysis. Each section was then placed in a separate sterile Eppendorf tube, placed on dry ice, and sent for high-throughput sequencing analysis (12 samples for each group, except for group 5, which contained 11 samples only; Appendix A). The colon samples were obtained and saved in the same way (12 samples for each group; Appendix A).

### 4.4. High-Throughput Sequencing Analysis and Reconstruction of Intestinal Microbiota Metabolic Activity

The “Multiomics Technologies of Living Systems” Scientific Research Laboratory (Kazan, Russia) performed the microbiota analysis. Genomic DNA was isolated from the contents of the mouse intestine using the FastDNA TM Spin Kit for Feces (MP Biomedicals, Santa Ana, CA, USA). Using specific primers, the bacterial 16S rRNA gene’s V3–V4 region was amplified (refer to Appendix A). During the second round of PCR amplification, each sample was barcoded using index primers after the AMPure XP bead-based PCR product (Beckman Coulter, Brea, CA, USA, CB55766755) was purified. Qubit dsDNA High Sensitivity Assay Kit (Invitrogen, Carlsbad, CA, USA) and Qubit 2.0 fluorometer (Invitrogen, Carlsbad, CA, USA) were used to determine the amplicon concentration. Using a Qubit 2.0 fluorometer and the Qubit dsDNA High Sensitivity Assay Kit (Invitrogen, Carlsbad, CA, USA), the concentration of amplicons was determined. To complete library preparation, the materials were mixed in an equal mole ratio prior to sequencing. Subsequently, the libraries were subjected to high-throughput sequencing (2 × 300 bp reads) (Illumina MiSeq, Illumina, CA, USA). The raw reads were processed using QIIME2 v2023.7.0 [39] and PICRUSt2 v2.5.2 software https://huttenhower.sph.harvard.edu/picrust/ (accessed on 12 September 2023). The microbial metabolic pathways encoded by the identified bacterial genomes were evaluated according to the sequencing data findings obtained with the PICRUSt2 v2.5.2 program. The most abundant pathways were identified by multiple t test analysis.

### 4.5. Statistical Data Analysis

The statistical software GraphPad Prism 10 v10.0.2 (171) was applied to process the data using nonparametric statistics techniques. Welch’s one-way analysis of variance (ANOVA) or multiple Mann–Whitney tests with the two-stage step-up approach (Benjamini, Krieger, and Yekutieli) (false discovery rate Q = 5%) were used to examine all in vivo experimental data. The standard deviation and mean were utilized to represent all data. *p* values less than 0.05 were considered to indicate statistical significance (* *p* < 0.05, ** *p* < 0.01, *** *p* < 0.001, **** *p* < 0.0001).

## Figures and Tables

**Figure 1 ijms-25-06611-f001:**
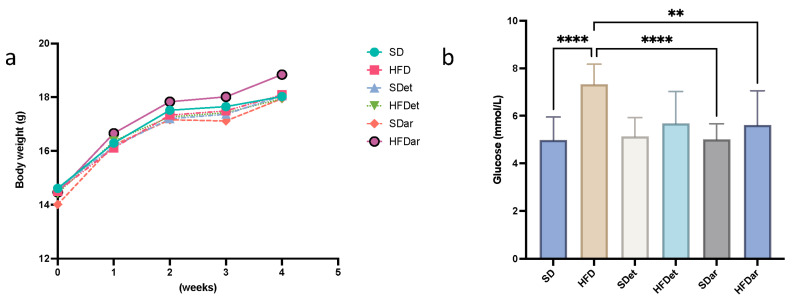
Dynamics of mouse BW and serum glucose levels in different study groups: (**a**) BW dynamic curves describing mouse weight changes during the 4-week feeding experiment; no significant differences in BW between study groups were found. (**b**) Results of serum glucose levels measured in the different experimental groups of mice; comparisons were carried out using one-way ANOVA followed by Tukey’s multiple comparison test (** *p* < 0.01, **** *p* < 0.0001). SD, standard diet; HFD, high-fat diet; SDet, standard diet with solvent addition; HFDet, high-fat diet with solvent addition; SDar, standard diet with C15 supplementation; HFDar, high-fat diet with C15 supplementation.

**Figure 2 ijms-25-06611-f002:**
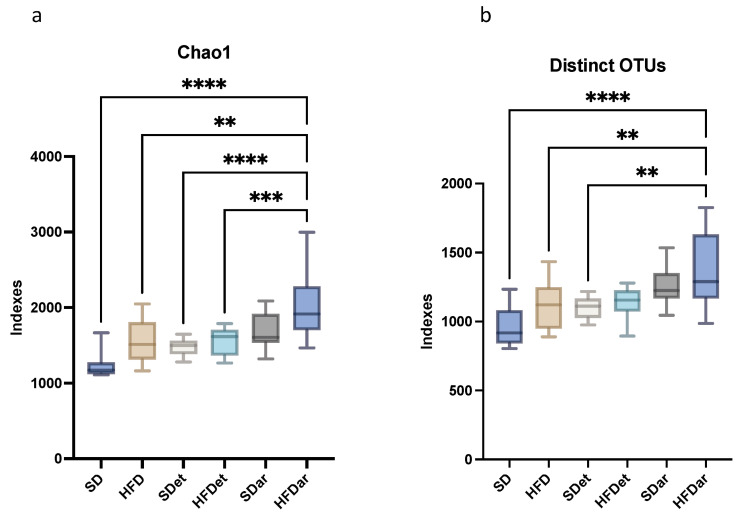
Results of one-way ANOVA followed by Tukey’s multiple comparison test for alpha diversity indices ((**a**)—for Chao1, (**b**)—for distinct OTUs), indicating changes in the richness of the mouse colon microbiota. ** *p* < 0.01, *** *p* < 0.001, **** *p* < 0.0001. SD, standard diet; HFD, high-fat diet; SDet, standard diet with solvent addition; HFDet, high-fat diet with solvent addition; SDar, standard diet with C15 supplementation; HFDar, high-fat diet with C15 supplementation.

**Figure 3 ijms-25-06611-f003:**
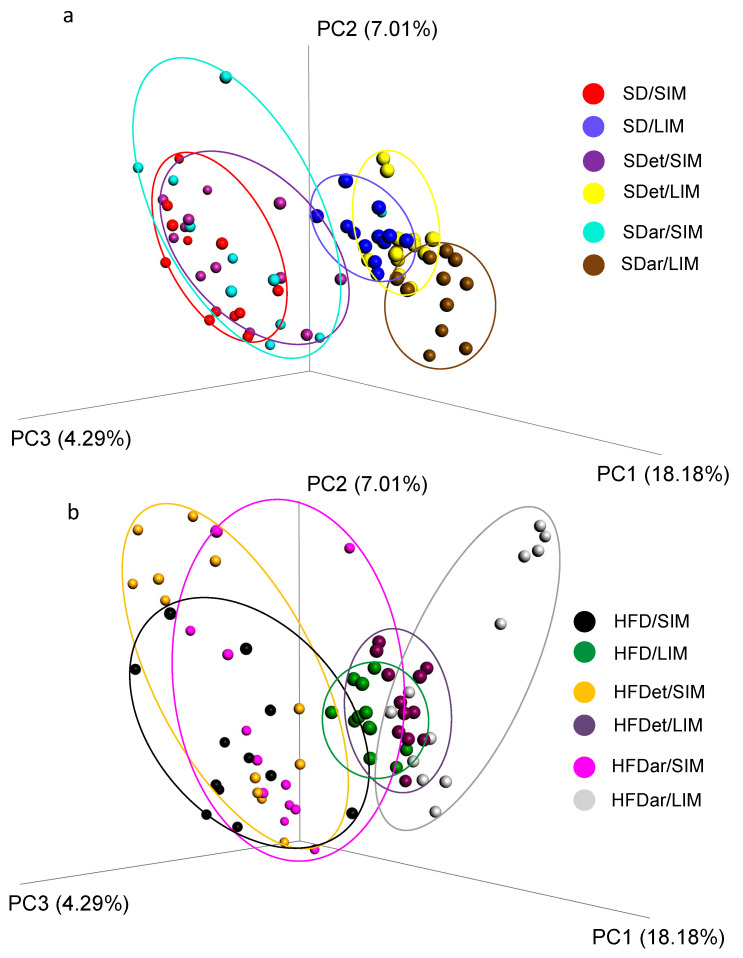
PCoA based on unweighted UniFrac distances evaluated for standard diet feeding groups (**a**) and high-fat diet feeding groups (**b**).

**Figure 4 ijms-25-06611-f004:**
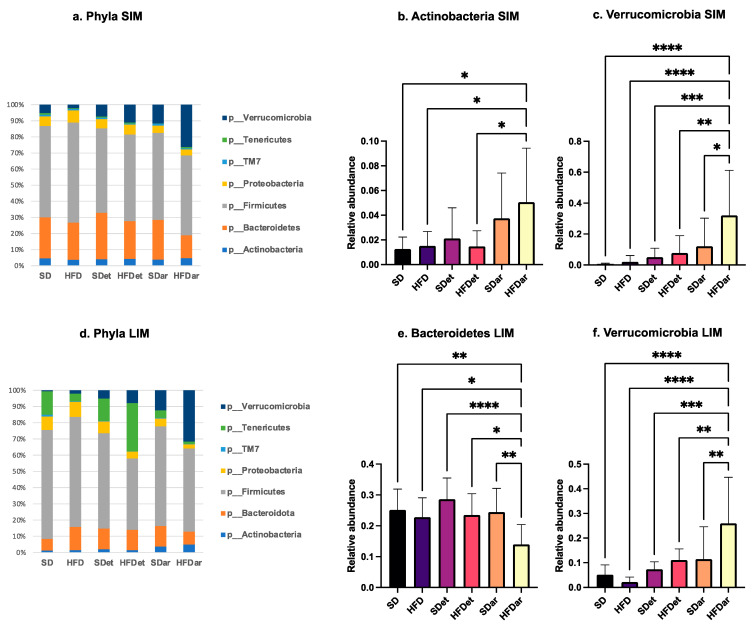
Effects of different diet types on the mouse SIM and LIM communities. (**a**) SIM phylum level, (**b**) *Actinobacteria* in SIM, (**c**) *Verrucomicrobia* in SIM, (**d**) LIM phylum level, (**e**) *Bacteroidota* in LIM, and (**f**) *Verrucomicrobia* in LIM. * *p* < 0.05, ** *p* < 0.01, *** *p* < 0.001, **** *p* < 0.0001. Comparisons were performed by one-way analysis of variance followed by Tukey’s multiple comparison test. SD, standard diet; HFD, high-fat diet; SDet, standard diet with solvent addition; HFDet, high-fat diet with solvent addition; SDar, standard diet with C15 supplementation; HFDar, high-fat diet with C15 supplementation.

**Figure 5 ijms-25-06611-f005:**
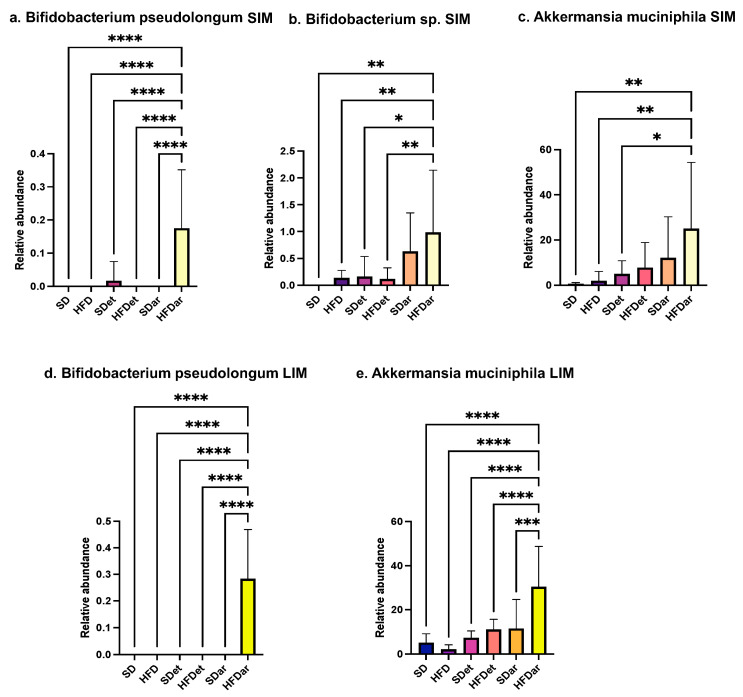
Effects of different diet types on mouse SIM and LIM communities at the species level. (**a**) *Bifidobacterium pseudolongum* in SIM; (**b**) *Bifidobacterium* sp. (other) in SIM; (**c**) *Akkermansia muciniphila* in SIM; (**d**) *Bifidobacterium pseudolongum* in LIM; and (**e**) *Akkermansia muciniphila* in LIM. * *p* < 0.05, ** *p* < 0.01, *** *p* < 0.001, **** *p* < 0.0001 for ANOVA. Comparisons were performed by one-way analysis of variance followed by Tukey’s multiple comparison test. SD, standard diet; HFD, high-fat diet; SDet, standard diet with solvent addition; HFDet, high-fat diet with solvent addition; SDar, standard diet with C15 supplementation; HFDar, high-fat diet with C15 supplementation.

**Table 1 ijms-25-06611-t001:** Alpha diversity index information. Values are expressed as means and standard deviations (StDev). The asterisk (*) indicates the presence of differences in the indexes compared to those of the control groups. All statistical data were analyzed using one-way analysis of variance (ANOVA) followed by Tukey’s multiple comparison test.

	SD		HFD		SDet		HFDet		SDar		HFDar	
	Mean	StDev	Mean	StDev	Mean	StDev	Mean	StDev	Mean	StDev	Mean	StDev
Jejunum												
Distinct OTUs	846.00	219.18	1004.58	154.91	968.58	202.39	954.08	262.23	1076.09	237.01	1248.00	404.42
Shannon Entropy	6.16	1.63	6.67	0.49	6.44	0.99	6.11	1.24	6.36	1.19	6.65	0.96
Berger–Parker Dominance	0.17	0.20	0.09	0.04	0.14	0.09	0.16	0.13	0.15	0.13	0.13	0.06
Chao1 Richness	1209.52	287.42	1407.98	248.51	1342.13	227.88	1353.98	296.85	1526.96	382.74	1856.23	632.63
Simpson Index	0.92	0.14	0.97	0.01	0.95	0.04	0.94	0.08	0.94	0.07	0.96	0.03
Inverse Simpson Index	1.13	0.28	1.03	0.01	1.05	0.05	1.08	0.11	1.06	0.09	1.05	0.04
Gini–Simpson Index	0.08	0.14	0.03	0.01	0.05	0.04	0.06	0.08	0.06	0.07	0.04	0.03
Colon												
Distinct OTUs *	1109.33	318.61	1123.83	174.48	1099.58	77.38	1136.08	114.13	1248.50	145.17	1365.00 *	262.00
Shannon Entropy	7.25	0.50	7.07	0.24	7.12	0.32	7.00	0.34	7.12	0.52	7.12	0.45
Berger–Parker Dominance	0.07	0.02	0.09	0.03	0.10	0.04	0.09	0.02	0.11	0.05	0.10	0.03
Chao1 Richness *	1503.54	473.34	1558.98	277.30	1476.15	115.74	1553.27	181.44	1673.63	253.43	2018.356 *	433.85
Simpson Index	0.98	0.01	0.98	0.01	0.98	0.01	0.97	0.01	0.97	0.01	0.97	0.01
Inverse Simpson Index	1.02	0.01	1.02	0.01	1.02	0.01	1.03	0.01	1.03	0.02	1.03	0.01
Gini–Simpson Index	0.02	0.01	0.02	0.01	0.02	0.01	0.03	0.01	0.03	0.01	0.03	0.01

**Table 2 ijms-25-06611-t002:** Differences in the representation of the metabolic pathways of the SI microbiota of C57BL/6 mice fed a standard diet compared to those fed a high fat diet supplemented with C15 (HFDar) according to multiple Mann–Whitney tests. The Q value reflects a false discovery rate of 5%. The mean rank difference values reflect the direction of changes in the abundance of metabolic pathways (values less than zero indicate an increased predicted representation of pathways, while values greater than zero indicate a decreased representation of pathways in the microbiota of mice fed a diet supplemented with C15 (red font)). Statistically significant values (*p* < 0.01) are indicated.

SD vs. HFDar (SIM)			
	*p* Value	Mean Rank Diff.	q Value
ubiquinol-7 biosynthesis (prokaryotic)	0.000009	11.33	0.000523
ubiquinol-9 biosynthesis (prokaryotic)	0.000009	11.33	0.000523
ubiquinol-10 biosynthesis (prokaryotic)	0.000009	11.33	0.000523
superpathway of L-phenylalanine biosynthesis	0.000009	−11.33	0.000523
superpathway of L-tyrosine biosynthesis	0.000009	−11.33	0.000523
ubiquinol-8 biosynthesis (prokaryotic)	0.000009	11.33	0.000523
superpathway of ubiquinol-8 biosynthesis (prokaryotic)	0.000009	11.33	0.000523
heme biosynthesis I (aerobic)	0.000033	10.83	0.001717
methanogenesis from acetate	0.000144	−10.17	0.00541
heterolactic fermentation	0.000144	10.17	0.00541
superpathay of heme biosynthesis from glutamate	0.000144	10.17	0.00541
Bifidobacterium shunt	0.000371	9.667	0.012028
polyisoprenoid biosynthesis (*E. coli*)	0.000496	9.5	0.012028
peptidoglycan biosynthesis II (staphylococci)	0.000496	−9.5	0.012028
superpathway of glucose and xylose degradation	0.000496	9.5	0.012028
pyrimidine deoxyribonucleotides de novo biosynthesis I	0.000496	9.5	0.012028
mevalonate pathway I	0.000496	9.5	0.012028
NAD salvage pathway II	0.000656	9.333	0.012305
superpathway of demethylmenaquinol-6 biosynthesis I	0.000656	−9.333	0.012305
superpathway of demethylmenaquinol-9 biosynthesis	0.000656	−9.333	0.012305
superpathway of geranylgeranyldiphosphate biosynthesis I (via mevalonate)	0.000656	9.333	0.012305
peptidoglycan biosynthesis IV (*Enterococcus faecium*)	0.000656	9.333	0.012305
superpathway of (Kdo)2-lipid A biosynthesis	0.001115	9	0.019997
L-1,2-propanediol degradation	0.001433	8.833	0.022737
pyrimidine deoxyribonucleotides de novo biosynthesis II	0.001433	8.833	0.022737
superpathway of pyrimidine deoxyribonucleotides de novo biosynthesis (*E. coli*)	0.001433	8.833	0.022737
superpathway of N-acetylneuraminate degradation	0.002316	8.5	0.031863
superpathway of menaquinol-9 biosynthesis	0.002316	−8.5	0.031863
superpathway of menaquinol-6 biosynthesis I	0.002316	−8.5	0.031863
superpathway of menaquinol-10 biosynthesis	0.002316	−8.5	0.031863
L-lysine biosynthesis II	0.002914	8.333	0.033402
superpathway of heme biosynthesis from glycine	0.002914	8.333	0.033402
superpathway of pyrimidine ribonucleosides salvage	0.002914	8.333	0.033402
superpathway of guanosine nucleotides de novo biosynthesis I	0.002914	8.333	0.033402
peptidoglycan maturation (meso-diaminopimelate containing)	0.002914	8.333	0.033402
nitrate reduction VI (assimilatory)	0.002914	−8.333	0.033402
reductive acetyl coenzyme A pathway	0.003637	−8.167	0.03573
enterobacterial common antigen biosynthesis	0.003637	8.167	0.03573
gluconeogenesis I	0.003637	8.167	0.03573
superpathway of glycerol degradation to 1,3-propanediol	0.003637	8.167	0.03573
acetylene degradation	0.003637	8.167	0.03573
pyrimidine deoxyribonucleotide phosphorylation	0.003637	8.167	0.03573
fatty acid elongation—saturated	0.004513	8	0.041385
ppGpp biosynthesis	0.004513	8	0.041385
pyruvate fermentation to acetate and lactate II	0.004513	8	0.041385
pyruvate fermentation to propanoate I	0.00556	−7.833	0.048812
superpathway of guanosine nucleotides de novo biosynthesis II	0.00556	7.833	0.048812

**Table 3 ijms-25-06611-t003:** Differences in the representation of the SI microbiota metabolic pathways of C57BL/6 mice fed a high fat diet compared to those fed a high fat diet supplemented with C15 (HFDar) according to multiple Mann–Whitney tests. The Q value reflects a false discovery rate of 5%. The mean rank difference values reflect the direction of changes in the abundance of metabolic pathways (values less than zero indicate an increased predicted representation of pathways, while values greater than zero indicate a decreased representation of pathways in the microbiota of mice fed a diet supplemented with C15 (red font)). Statistically significant values (*p* < 0.01) are indicated.

HFD vs. HFDar (SIM)			
	*p* Value	Mean Rank Diff.	q Value
superpathway of ubiquinol-8 biosynthesis (prokaryotic)	0.000033	10.83	0.004412
ubiquinol-7 biosynthesis (prokaryotic)	0.00005	10.67	0.004412
ubiquinol-9 biosynthesis (prokaryotic)	0.00005	10.67	0.004412
ubiquinol-10 biosynthesis (prokaryotic)	0.00005	10.67	0.004412
ubiquinol-8 biosynthesis (prokaryotic)	0.00005	10.67	0.004412
heme biosynthesis I (aerobic)	0.000274	9.833	0.016529
hexitol fermentation to lactate, formate, ethanol and acetate	0.000274	9.833	0.016529
superpathway of menaquinol-8 biosynthesis II	0.000371	9.667	0.016529
1,4-dihydroxy-6-naphthoate biosynthesis II	0.000371	9.667	0.016529
1,4-dihydroxy-6-naphthoate biosynthesis I	0.000371	9.667	0.016529
superpathway of L-alanine biosynthesis	0.000656	−9.333	0.026551
superpathway of sulfur oxidation (*Acidianus ambivalens*)	0.000858	9.167	0.027282
superpathay of heme biosynthesis from glutamate	0.000858	9.167	0.027282
L-1,2-propanediol degradation	0.000858	9.167	0.027282

**Table 4 ijms-25-06611-t004:** Differences in the representation of LI microbiota metabolic pathways of C57BL/6 mice fed a high fat diet compared to those fed a high fat diet supplemented with C15 (HFDar) according to multiple Mann–Whitney tests. The Q value reflects a false discovery rate of 5%. The mean rank difference values reflect the direction of changes in the abundance of metabolic pathways (values less than zero indicate an increased predicted representation of pathways, while values greater than zero indicate a decreased representation of pathways in the microbiota of mice fed a diet supplemented with C15 (red font)). Statistically significant values (*p* < 0.01) are indicated.

HFD vs. HFDar (LIM)			
	*p* Value	Mean Rank Diff.	q Value
glucose and glucose-1-phosphate degradation	<0.000001	−12	0.000031
1,4-dihydroxy-2-naphthoate biosynthesis I	<0.000001	−12	0.000031
superpathway of menaquinol-8 biosynthesis I	<0.000001	−12	0.000031
superpathway of menaquinol-7 biosynthesis	<0.000001	−12	0.000031
superpathway of demethylmenaquinol-8 biosynthesis	<0.000001	−12	0.000031
superpathway of phylloquinol biosynthesis	<0.000001	−12	0.000031
superpathway of menaquinol-11 biosynthesis	<0.000001	−12	0.000031
superpathway of menaquinol-12 biosynthesis	<0.000001	−12	0.000031
superpathway of menaquinol-13 biosynthesis	<0.000001	−12	0.000031
superpathway of L-alanine biosynthesis	<0.000001	−12	0.000031
sulfate reduction I (assimilatory)	0.000001	−11.83	0.000051
superpathway of sulfate assimilation and cysteine biosynthesis	0.000001	−11.83	0.000051
superpathway of L-methionine biosynthesis (by sulfhydrylation)	0.000005	−11.5	0.000165
fucose degradation	0.000014	−11.17	0.000417
superpathway of pyridoxal 5′-phosphate biosynthesis and salvage	0.000022	−11	0.000614
superpathway of 2,3-butanediol biosynthesis	0.000033	10.83	0.000813
1,4-dihydroxy-6-naphthoate biosynthesis I	0.000033	10.83	0.000813
TCA cycle VI (obligate autotrophs)	0.00005	−10.67	0.001083
1,4-dihydroxy-6-naphthoate biosynthesis II	0.00005	10.67	0.001083
preQ0 biosynthesis	0.000072	−10.5	0.001489
adenosylcobalamin salvage from cobinamide I	0.000144	−10.17	0.002603
L-histidine degradation I	0.000144	10.17	0.002603
pyridoxal 5′-phosphate biosynthesis I	0.000144	−10.17	0.002603
superpathway of (Kdo)2-lipid A biosynthesis	0.000273	9.833	0.004556
superpathway of menaquinol-8 biosynthesis II	0.000274	9.833	0.004556
glucose degradation (oxidative)	0.000337	−9	0.005373
superpathway of UDP-N-acetylglucosamine-derived O-antigen building blocks biosynthesis	0.000496	9.5	0.007619
L-lysine biosynthesis II	0.000656	9.333	0.009726

## Data Availability

The data presented in this study are available upon request from the corresponding author.

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
