# Peer review of "Supplementation of a High-Fat Diet with Pentadecylresorcinol Increases the Representation of Akkermansia muciniphila in the Mouse Small and Large Intestines and May Protect against Complications Caused by Imbalanced Nutrition"

_ijms, 2024, doi:10.3390/ijms25126611_

Round 1

Reviewer 1 Report

Comments and Suggestions for Authors

General comments

The manuscript contains the study about a diet high in pentadecylresorcinol for 28 days. The manuscript investigated separately the effects on the microbiota of the small and large intestine, which is unusual, and is nowadays difficult to perform in many countries as it requires a large number of experimental animals (in this manuscript, it were used 72). Overall, the manuscript is well written and structured (with the odd exception of putting the discussion before the materials and methods). The science developed is good. However, I believe that the manuscript could be improved if the authors correct the following aspects that I detail as specific comments:

Specific comments:

The similarity index with the previously published manuscript entitled “Alkylresorcinols as New Modulators of the Metabolic Activity of the Gut Microbiota” (23%) should be decreased until the final publication.

Page 2, lines 51-53: Additionally to “metabolic protectors”, a diet or a transplantation from other donor can contains other important factors that modulate gut microbiota. Especially, in recent years the influence of virome in metabolic diseases via gut microbiota modulation was pointed out.

Page 2, line 57 “GLP” should be define the first time that appears in the manuscript.

Page 3, figure 1. I don´t understand the p levels stated. In the figure are not showed any asterisk, so why are in the footnote? And why are cited only differences revealed by two or four asterisks? Why about one and three? I do not understand it.

Page 3, line 103: Please, delete the end dot.

Page 4, Table 1. There is no need for so many decimal levels. Please reduce them to a maximum of 3-4 to make the table more manageable for the editor and more easily readable for the reader.

Tables 9-10. Figure 6. font size must be increased so that the text can be read correctly by the reader. If the software does not allow it, you may want to change the formatting to a table or other model that allows clear reading.

Page 10, lines 235-237: Please add the Ethics Committee approval code.

Page 10, line 258: Why were chosen a dosage of 0.006 mg/day for 28 days? Please, explain.

Page 11, line 275. In text headings, it is not convenient to express names by abbreviations, it is better to use the full name (LIM and SIM).

Why were stated Discussion section after Materials and Methods section? This is not the usual format of the journal…Discussion section should be placed after the results section.

In references 5, 23 and 32, the names of the journals were cited by their full name and not by the abbreviated name

Reviewer 2 Report

Comments and Suggestions for Authors

The paper by Zabolotneva describes the impact of AR supplementation on the composition of the jejunum and colon microbiota and metabolic activity under conditions of a short-term high-fat diet.

Although the work is interesting, there are several fundamental concerns that should be addressed. The main objection is the lack of certain methods in addition to the methodology described. In particular, the authors should provide some functional analyses, such as Western blot analyses of target enzymes or the activity of specific enzymes relevant to metabolic pathways affected by diet. The integration of additional low-throughput methods such as qPCR, targeted metabolomics, enzyme activity assays and histological analyses can provide a more comprehensive understanding of the relationship between diet, microbiota and metabolism. These methods can improve the robustness of the results and increase the publishability of the work.

SPECIFIC COMMENTS

RESULTS

Lines 83, 85.what does value 14.4+/-0.96 and 18.15+/-1.14 g represent? Is this the value of one group or some other group? It is unclear.

Fig 1b. It appears that bars for each group on the graph are missing along with marks of significances. Please correct this. Also, please mark statistical significances described in the legends to figures (i.e. p less than 0.01 or 0.001)

Lines 121, versus 123. If the authors perform experiments that accompany 6 groups, then they should use the same statistical tests for multiple comparisons, i.e. if oneway anova is more appropriate for comparisons between more than 2 groups (and it is indeed), then the authors should avoid t-test which is usually used for comparisons of differences between two groups.

Figure 2. Again it appears that bars are missing. If the authors decided to show only error bars, please incorporate bars for each group, as this presentation is quite unusual and confusing.

Line 136: the statement "did not reveal a significant difference" is incorrect given the P value, as P value shows that There is a statistically significant difference between the bacterial communities of SIM and LIM, as indicated by the P value (P = 0.001). However, the R value of 0.2527 suggests that the magnitude of this difference is small. Please correct this statement.

Figure 5. Please adjust the y-axis on the graphs so that the bars of all groups are visible. You can easily do that in graphpad.

Line 206. what does the expression „Bifidobacte cum shunt“ mean? Please clarify.

Figure 6. please adjust font on volcano plots so the names could be visible. It is very hard to read it in a such a small font. Also, it appears that there are no any values in these plots, only names. Please correct this.

GENERAL COMMENT FOR FIGURES: please equalize fonts and formats of all graphs!

Please check the order of the sections in the paper. Do the materials and methods go before the discussion and after the results?

Comments on the Quality of English Language

minor editing required.

Round 2

Reviewer 2 Report

Comments and Suggestions for Authors

The authors have adequately addressed my concerns, and I now recommend the paper for publication. However, I must emphasize the importance of considering the inclusion of other methods in future research, beyond just metabolomic analysis, to provide a more comprehensive understanding. Despite the authors' justifications for the current limitations, incorporating diverse methods will significantly enhance the depth and breadth of future studies."

MINOR:

"The bars in the graphs are still not visible in the PDF version of the document. Please ensure that the bars are clearly visible in the final version to be published."
